# A pilot randomised trial of a brief virtual reality scenario in smokers unmotivated to quit: Assessing the feasibility of recruitment

Olga Perski [1,2]*, Trupti Jambharunkar [1,2], Jamie Brown [1,2], Dimitra Kale [1,2]

1 Department of Behavioural Science and Health, University College London, London, United Kingdom,
2 SPECTRUM Consortium, London, United Kingdom

* olga.perski@ucl.ac.uk

## Abstract

Individual-level interventions for smokers unmotivated to quit remain scarce and have had limited success. Little is known about the potential of virtual reality (VR) for delivering messaging to smokers unmotivated to quit. This pilot trial aimed to assess the feasibility of recruitment and acceptability of a brief, theory-informed VR scenario and estimate proximal quitting outcomes. Unmotivated smokers (recruited between February-August 2021) aged 18+ years who had access to, or were willing to receive via post, a VR headset were randomly assigned (1:1) using block randomisation to view the intervention (i.e., a hospital-based scenario with motivational stop smoking messaging) or a 'sham' VR scenario (i.e., a scenario about the human body without any smoking-specific messaging) with a researcher present via teleconferencing software. The primary outcome was feasibility of recruitment (i.e., achieving the target sample size of 60 participants within 3 months of recruitment). Secondary outcomes included acceptability (i.e., positive affective and cognitive attitudes), quitting self-efficacy and intention to stop smoking (i.e., clicking on a weblink with additional stop smoking information). We report point estimates and 95% confidence intervals (CIs). The study protocol was pre-registered (osf.io/95tus). A total of 60 participants were randomised within 6 months (intervention: n = 30; control: n = 30), 37 of whom were recruited within a 2-month period of active recruitment following an amendment to gift inexpensive (£7) cardboard VR headsets via post. The mean (SD) age of participants was 34.4 (12.1) years, with 46.7% identifying as female. The mean (SD) cigarettes smoked per day was 9.8 (7.2). The intervention (86.7%, 95% CI = 69.3%-96.2%) and control (93.3%, 95% CI = 77.9%-99.2%) scenarios were rated as acceptable. Quitting self-efficacy and intention to stop smoking in the intervention (13.3%, 95% CI = 3.7%-30.7%; 3.3%, 95% CI = 0.1%-17.2%) and control (26.7%, 95% CI = 12.3%-45.9%; 0%, 95% CI = 0%-11.6%) arm were comparable. The target sample size was not achieved within the feasibility window; however, an amendment to gift inexpensive headsets via post appeared feasible. The brief VR scenario appeared acceptable to smokers unmotivated to quit.

**Data Availability Statement:** The data underpinning the analyses are openly available via Zenodo: https://doi.org/10.5281/zenodo.5747705.

The R code is openly available via GitHub: https://github.com/OlgaPerski/VR_study.

**Funding:** OP, DK, and TJ receive salary support from Cancer Research UK (C1417/A22962). JB, OP, DK and TJ are members of SPECTRUM – a UK Prevention Research Partnership Consortium (MR/S037519/1). The funders had no role in study design, data collection and analysis, decision to publish, or preparation of the manuscript.

**Competing interests:** I have read the journal's policy and the authors of this manuscript have the following competing interests: JB has received unrestricted research funding unrelated to the current study from Pfizer and Johnson & Johnson.

## Author summary

Virtual reality (VR)–i.e., the creation of a digital environment which gives rise to strong sensory experiences–can be used to deliver motivational messaging to smokers unmotivated to stop. However, little is currently known about the feasibility of recruiting smokers with access to a VR headset into an online trial or the acceptability of a VR scenario focused on the health consequences of smoking in smokers unmotivated to stop. We helped develop a brief, theory-based VR scenario in which participants attended a consultation with a chest physician in a hospital clinic room. We randomised 60 adult smokers to view the intervention or control VR scenario. We found that although the target sample size was not achieved within 3 months from the trial start date (i.e., the pre-specified feasibility window), gifting inexpensive VR headsets to smokers via post appeared feasible. In addition, the brief VR scenario appeared acceptable to smokers unmotivated to stop. This study was–to our knowledge–the first pilot randomised trial of a VR scenario designed specifically for adult smokers unmotivated to stop. Future work would benefit from optimising the VR scenario through, for example, think aloud methodology and then proceed to a larger-scale evaluation study.

## 1. Introduction

Smoking is a leading cause of premature ill-health and death, directly responsible for >7 million global deaths per year [1]. Supporting smokers to quit is a public health priority. Motivation to stop smoking is a key predictor of quit attempts, with approximately 80–85% of smokers in England characterised as not highly motivated (i.e., not expressing an intention to stop in the next three months) [2,3]. Although population-level mass media campaigns have been found to successfully prompt quit attempts [4], individual-level interventions designed specifically for smokers unmotivated to quit remain scarce [5]. Available individual-level interventions for smokers unmotivated to quit via, for example, video messaging have had limited success in prompting quit attempts [6]. Virtual reality (VR) may offer an innovative and immersive means of delivering more salient stop smoking messaging to people unmotivated to quit, with a view to prompting quit attempts that would otherwise not occur. We therefore conducted a pilot randomised trial to i) assess the feasibility of recruitment and acceptability of a brief, theory-informed VR scenario and ii) estimate proximal quitting outcomes in smokers unmotivated to quit.

VR refers to the creation of a digital environment that has the potential to evoke strong sensory experiences, accessed through increasingly widespread technologies such as head-mounted displays [7]. Although feelings of immersion (e.g., transportation to a different, virtual environment, losing track of time) are characteristic of VR more broadly, the level of immersion experienced by users differs depending on the specific technology used, the number of senses engaged (e.g., vision, hearing, smell) and the level of interactivity (e.g., whether users can interact with objects in the virtual environment through, for example, a joystick) [8]. The potential of VR for delivering interventions to people with mental health problems, including substance use, has received ample attention in the last decade [7,9–16]. However, few clinical trials have been conducted, and VR deployments for substance use–including smoking cessation–tend to have focused on the delivery of cue exposure therapy. For example, a meta-analysis of 18 studies with 541 smokers found that VR environments in which participants were exposed to smoking-related cues (as compared with scenarios without such cues) reliably induced smoking urges [14]. However, few studies have explored the potential of VR

for delivering motivational messaging to smokers unmotivated to quit. A feasibility and pilot study explored the effect of a motivational VR scenario focused on the progression of smoking-related disease in smokers unmotivated to stop; however, this study was limited to young adult smokers who completed the testing within a laboratory setting [17].

We therefore helped develop an immersive, brief VR scenario–suited for delivery in smokers' own homes and accessible via commercially available head-mounted displays without involving a joystick–grounded in the Extended Parallel Process Model (EPPM) [18]. The EPPM proposes that the effect of threatening health messages (e.g., information or communications highlighting that smoking causes lung cancer) on people's motivation to protect themselves against said threat (e.g., through behaviour change) depends on two things: 1) people's emotional responses to the message, and 2) their response- and self-efficacy beliefs (i.e., beliefs that the suggested action to avoid the threat is effective and that one has the ability to act). Previous research guided by the EPPM involving pictorial health warning labels on cigarette packages has found an association of greater self-efficacy and stronger emotional responses to health warning labels with prospective quit attempts [19]. We therefore developed a hospital-based VR scenario, in which participants are invited to attend a consultation with a chest physician in a hospital clinic room and are told that they have had an abnormal chest scan which requires follow-up (i.e., a threatening health message). They are subsequently presented with a brief, written message to boost their response- and quitting self-efficacy. The hypothesised mechanism of action of the EPPM-based scenario is increased susceptibility to smoking-related diseases (i.e., a health threat) combined with increased response- and quitting self-efficacy, which are jointly expected to increase participants' intention to stop smoking. With regards to the level of immersion, users are not able to interact with the physician via, for example, a joystick and the primary senses engaged are vision and hearing (but not smell, taste or haptics).

Following user testing but prior to conducting a large-scale randomised controlled trial, we considered it important to conduct a pilot trial to assess the feasibility of recruitment and acceptability of the VR scenario. Acceptability of digital health interventions is a multifaceted concept, with available frameworks converging on the view that acceptability captures people's affective and cognitive attitudes towards a given digital health intervention [20,21]. Specifically, this pilot randomised trial aimed to address the following research questions (RQs):

RQ1: Is it feasible to recruit smokers unmotivated to quit with access to a VR headset to take part in a randomised trial?

RQ2: In smokers unmotivated to stop, is a VR scenario that emphasises the health consequences of smoking perceived as acceptable?

RQ3: Does an active, compared with a 'sham', VR scenario lead to greater i) perceived susceptibility to smoking-related diseases; ii) perceived response-efficacy; iii) perceived quitting self-efficacy; and iv) intention to stop smoking (as measured with a behavioural indicator)?

## 2. Methods

### 2.1 Study design

The CONSORT checklist of pilot trials [22] was used to inform the study design and write-up (see S1 CONSORT Checklist). This was a parallel-arm, pilot randomised trial conducted remotely with unmotivated, adult smokers randomised to the intervention and control arms in a 1:1 ratio using block randomisation (block size = 5). The random sequence was generated by the first author in R v.3.6.3 using the *blockrand* package [23]. According to the National

Institute for Health Research, pilot studies are "a smaller version of the main study used to test whether the components of the main study can all work together" [24]. The study protocol and analysis plan were pre-registered on the Open Science Framework (osf.io/95tus). We aimed to recruit a total of 60 participants (30 in each arm). For pilot studies, sample sizes between 24 and 50 participants have been recommended [25–27]. We aimed to recruit an additional 10 participants to account for potential study dropout. This study was single blinded (i.e., participants were not aware of the group allocation).

## 2.2 Eligibility criteria

**2.2.1 Inclusion criteria.** Participants were eligible to take part if they: i) were aged 18 + years; ii) were a fluent English speaker; iii) were a daily or non-daily smoker; iii) were considered 'unmotivated' to stop smoking in the next three months (i.e., a Motivation To Stop Scale (MTSS) score of ≤5; [3,28]); iv) had access to a VR headset capable of running the YouTube app (without or with the help of a smartphone); v) were willing and able to meet with a researcher to complete the online testing via teleconferencing software (e.g., Microsoft Teams, Zoom); and vi) had corrected-to-normal vision.

**2.2.2 Amendments to the inclusion criteria.** Due to a slower than expected recruitment rate, the inclusion criteria were amended in April 2021 (registered on the OSF prior to implementation; osf.io/5xzru/) to no longer restrict recruitment to participants with access to a VR headset. Instead, participants residing in the United Kingdom (UK) who were willing to be gifted an inexpensive headset (i.e., Google Cardboard worth ~£7) via post were also eligible to take part. The geographical limit was set to reduce postal costs, reduce delay between consenting and receipt of the headset, and minimise the testing burden caused by time differences between researchers and participants in different time zones.

**2.2.3 Exclusion criteria.** Given the nature of the hospital-based VR scenario, which aimed to increase smokers' perceived susceptibility to cancer (described in detail in section 2.5 below), we did not judge it ethical to include participants with a cancer diagnosis. Therefore, these participants were not eligible to take part, assessed by asking: "Have you been diagnosed with cancer?". For a similar reason, participants with severe health anxiety were not eligible, determined by a score of ≥5 on the validated 14-item Whitely Index [29,30].

## 2.3 Sample recruitment

Participants were recruited by adverts stating that the researchers were looking for smokers to provide feedback on a brief VR scenario, which mentioned a gift voucher to reimburse participants for their time. Adverts were placed on social media (i.e., Twitter, Facebook, Reddit), Prolific (www.prolific.co), e-mails sent through university mailing lists, and the researchers' networks.

## 2.4 Measures and procedure

Interested participants were asked to complete screening questions relating to the above eligibility criteria via a brief online survey, hosted by Qualtrics. They were also asked to provide information about their i) country of residence, ii) gender (male, female, in another way), iii) occupational status (manual, non-manual, other), iv) whether they owned a VR headset (no, yes), v) type of VR headset (free text response), vi) experience with VR headsets (none, limited, some, substantial), vii) cigarettes smoked per day (CPD), viii) time to first cigarette (TTFC), ix) whether they had made any serious attempts to stop smoking in the past year, and x) whether they had ever used one or more behavioural and/or pharmacological stop smoking aids from a list of options (i.e., nicotine replacement therapy, varenicline, bupropion, e-

cigarettes, group counselling, individual counselling, telephone helpline, written materials, website, app).

In the first recruitment phase (prior to the amendment to send headsets), eligible participants were randomised via Qualtrics' block randomisation function to the intervention or control arms. They were subsequently invited via e-mail to complete the online testing with a researcher present via videoconferencing software. When booking the meeting, participants received instructions to have their smartphone and/or VR headset charged and ready for the online testing. Instructions were identical for intervention and control participants. Reminders to book a meeting were sent to control and intervention participants with equal frequency (i.e., two reminders).

In the second recruitment phase (following the amendment to send headsets), participants who indicated that they did not have access to a VR headset were asked to meet with the researcher twice: first to provide their personal details (i.e., name, postal address, contact number) to enable the researcher to order a VR headset via Amazon to be delivered to the participant, and second to complete the online testing. After having booked a meeting to complete the online testing (irrespective of whether participants were sent a headset), participants were randomised to the intervention or control arms.

During the online testing, participants allocated to the intervention arm were asked to search for and view the active VR scenario within the YouTube app. Participants allocated to the control arm were asked to search for and view the 'sham' scenario. Immediately after viewing the VR scenario, participants were asked to complete a post-test survey, hosted by Qualtrics.

First, acceptability was examined by assessing participants' affective and cognitive attitudes towards the respective scenarios [21]. As both the intervention and control scenarios were delivered in VR, we aimed to assess the acceptability of the specific content delivered in VR format (i.e., there were no 2D controls to examine the comparative acceptability of VR per se). Although there is no consensus definition of acceptability, we recently proposed that acceptability may usefully be considered an emergent property of a complex, adaptive system of multiple, interacting components (e.g., beliefs, knowledge), which is experienced by the individual as a gut reaction or sudden insight [20]. Until further questionnaire development and validation work has been conducted, we reasoned that this sudden insight may be usefully operationalised as affective and cognitive attitudes. Affective attitude was measured by asking participants to rate their agreement with the following statements on a 7-point Likert scale (i.e., 'strongly disagree' to 'strongly agree'): "The scenario made me feel angry"; "The scenario made me feel distressed". Cognitive attitude was measured by asking participants to rate their agreement with the following statement on a 7-point Likert scale (i.e., 'strongly disagree' to 'strongly agree'): "The scenario was useful to me". Responses were coded as 'acceptable' if participants selected 'disagree' or 'strongly disagree' for the affective attitude, and 'agree' or 'strongly agree' for the cognitive attitude, and 'unacceptable' otherwise. These items were developed for the purposes of the present study and have not been validated.

Second, participants were asked to provide a rating of their perceived susceptibility to smoking-related diseases, perceived response-efficacy and perceived quitting self-efficacy. Perceived susceptibility was measured with three items: "What do you think your likelihood is of developing (or if you have, the worsening of) the following diseases if you continue smoking?" i) cancer, ii) heart disease, and iii) lung disease. Each were assessed on a 5-point Likert scale (i.e., 'not at all' to 'very likely') [31]. Responses were coded as 'susceptible' ('moderately likely' or 'very likely') or 'not susceptible' (all other response options). Perceived response-efficacy was measured on a 5-point Likert scale ('strongly disagree' to 'strongly agree') with the following items: "Quitting smoking is an effective protector against cancer"; "Quitting smoking is an

effective protector against heart disease"; and "Quitting smoking is an effective protector against lung disease." Due to a coding error when setting up the Qualtrics survey, 7-point (rather than 5-point) Likert scales were used ('strongly disagree' to 'strongly agree'). Responses were coded as 'confident' ('agree' or 'strongly agree') or 'not confident' (all other response options). Perceived quitting self-efficacy was measured on a 5-point Likert scale (i.e., 'not at all' to 'extremely') with the following item: "How confident are you in your ability to abstain from smoking?" [32]. Responses were coded as 'confident' ('very' or 'extremely') or 'not confident' (all other response options).

Participants were then presented with a link to a website with additional, evidence-based information about smoking cessation (http://www.nhs.uk/smokefree). A record was made if a participant clicked on the link provided–the act of clicking was interpreted as a behavioural indicator of intention to stop smoking.

As compensation for their time, participants with their own headset received a £5 gift voucher (or equivalent payment via Prolific) upon completion of the study. Participants without a headset were gifted a cardboard headset (~£7).

## 2.5 Intervention

**2.5.1 Intervention–active VR scenario.**   The active VR scenario was informed by the Extended Parallel Process Model [18] and was developed by an independent software developer ('MasterChange') with input from the researchers. In the immersive scenario, participants are invited to attend a consultation with a chest physician in a hospital clinic room (see Fig 1). Participants are told by the chest physician that they have had an abnormal chest scan which requires follow-up. They are subsequently presented with a brief, written message to boost their response- and quitting self-efficacy (i.e., "Stopping smoking is the single most important thing that you can do to improve your health and quality of life. We now know that the best way to stop is the combination of support, advice and stop smoking treatments provided by your local stop smoking services. The expert service will greatly increase your chances of success and is available for free. Millions of people in England have stopped with their help. They were just like you. Believe in yourself and commit to quitting today!"). See S1 Data for the behaviour change techniques (BCTs) included in the intervention scenario, coded against

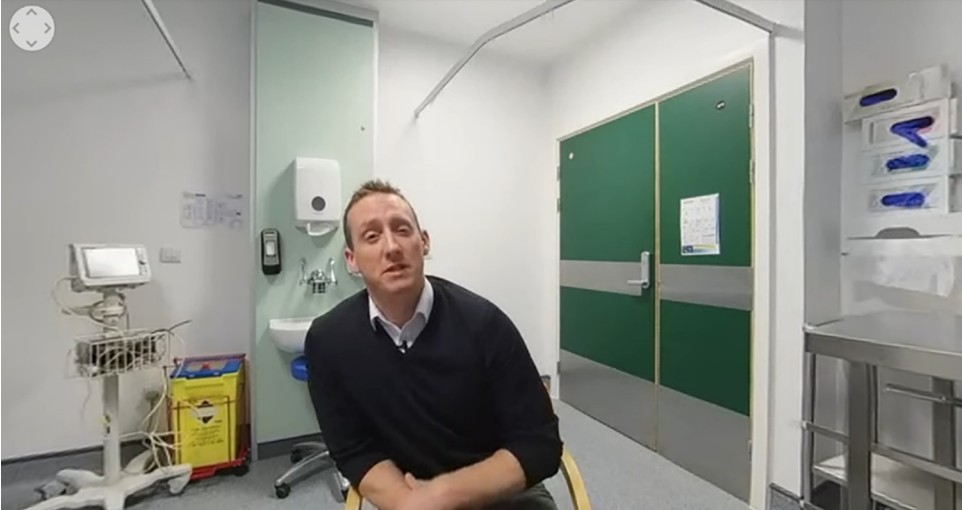

**Fig 1. Screenshot of the active VR scenario.** Reproduced with permission from MasterChange.

a 44-item taxonomy of BCTs used in behavioural smoking cessation interventions [33]. The hypothesised mechanism of action of the scenario is increased susceptibility to smoking-related diseases combined with increased response- and quitting self-efficacy, which are jointly expected to increase participants' intention to stop smoking. Evidence suggests that fear appeals tend to undermine behaviour change unless they are also accompanied by materials that boost participants' self-efficacy [34]. Due to the potential of VR to evoke strong sensory and emotional experiences–e.g., prior research shows it can reliably induce smoking urges [14]–we expected the immersive VR scenario to evoke strong emotional responses to the health message. Although we did not include a 2D video control, we expected emotional responses to be stronger than if, for example, viewing a 2D video with the same health message.

**2.5.2 Control–'sham' VR scenario.** Participants allocated to the control arm were asked to view an immersive VR scenario of equivalent length about the human body, developed by 'Hybrid Medical' (see Fig 2). The content of the scenario was neutrally framed. This was deemed a suitable control condition as it exposes participants to the same immersive VR environment for a similar length of time but without providing any smoking-specific messaging.

Formative user testing of the intervention and control scenarios and the study materials was conducted in December 2020. The feedback received was used to improve the study materials (i.e., enhancing the visibility of the weblink to the additional stop smoking information by increasing the font size).

## 2.6 Ethical approval

The study fell within the scope of 'The optimisation and implementation of interventions to change behaviours related to health and the environment', approved by UCL's Research Ethics Committee (Project ID: CEHP/2020/579). Prior to taking part, participants were asked to read an information sheet detailing the study procedures and provide electronic informed consent, including consent for their fully anonymised data to be shared on an open science repository.

## 2.7 Data analysis

The analyses were conducted in R v.3.6.3. Descriptive statistics (means, percentages) were calculated to describe the sample.

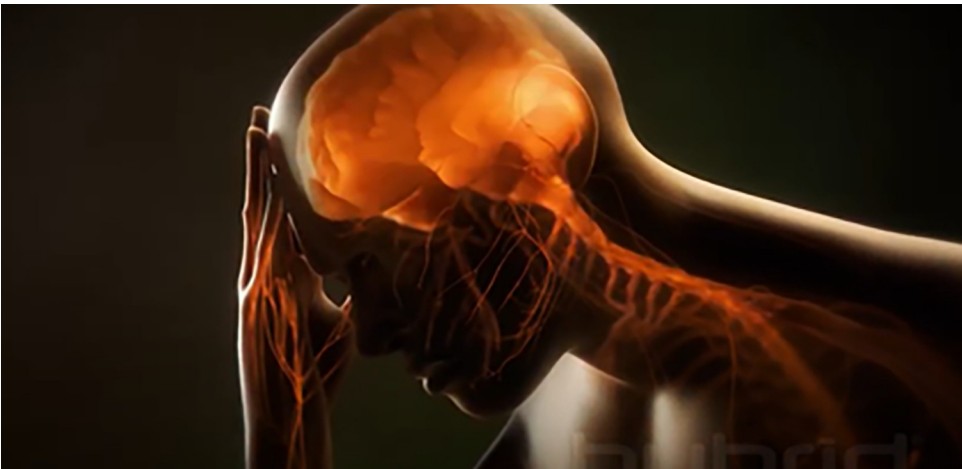

**Fig 2. Screenshot of the 'sham' VR scenario.** Reproduced with permission from Hybrid Medical.

To address RQ1, we considered a large-scale randomised controlled trial (RCT) of the brief VR scenario feasible to deliver if we were able to recruit the target sample size of 60 participants within three months from the trial start date (i.e., January 2021). The feasibility target was selected on the basis of the following assumptions/considerations: i) recruitment for a large-scale RCT would ideally be completed within two years, and ii) expecting a three-fold increase in the rate of recruitment with added resource (i.e., a full- as opposed to part-time researcher working on the project in addition to a budget of £2.50 per recruited participant via Facebook advertising, or a total of £4000). A power simulation in R ($n_{sims}$ = 500) indicated that a sample size of 1562 participants would provide >80% power (with one-tailed alpha set to 5%) to detect a projected 5% increase in quit attempts at a 6-month follow-up in the intervention arm compared with the 'sham' control (i.e., 25% vs. 20%, OR = 1.25). The anticipated effect size is based on what is known about the rate of unaided quit attempts in the past 6 months in England (www.smokinginengland.info)–removing highly motivated people–and a previous intervention for unmotivated smokers, which yielded a 9% intervention difference in quit attempts at 6 months [5]. As our brief VR scenario is judged to be less intensive than the intervention provided in Carpenter et al. (2011), we anticipate a slightly smaller intervention difference of 5%. Therefore, based on the above assumptions, being able to recruit 60 participants within three months from the trial start date (or ~200 participants with added resource), this would ensure we would be able to recruit ~1600 participants within 24 months from the trial start date.

### 2.8 Deviations from the pre-specified analysis plan

To address RQs 2–3, we had specified in the pre-registered analysis plan that linear and logistic regression analyses would be conducted to examine group differences in acceptability, perceived susceptibility to smoking-related diseases, perceived response-efficacy, perceived quitting self-efficacy, and intention to stop smoking. However, following statistical review but before completing data collection, we deemed it more appropriate to instead calculate point estimates and 95% confidence intervals (CIs) due to the small sample size, opting for a more descriptive approach. As a result of the peer review process, however, we also present results from linear regression analyses with the outcome variables (except for intention to stop smoking) operationalised as continuous in Table A in S1 Table. We also present the bivariate correlations between the acceptability and perceived susceptibility indicators in Table B in S1 Table.

## 3. Results

### 3.2 Participant characteristics

A total of 614 participants completed the screening, of whom 376 (61.2%) were eligible to take part. A total of 103 (16.8%) participants responded to the invitation to attend the online testing session and were randomised, with 62 attending the testing session and completing the study (10.1%). Due to a technical glitch, post-task survey data were lost for two participants (intervention: n = 1; control: n = 1), leaving an analytic sample of 60 participants (see Fig 3). The mean age of participants was 34.4 (12.1) years, with 46.7% identifying as female (see Table 1). Most participants were residing in the United Kingdom and half had a non-manual occupation.

### 3.3 Feasibility of recruitment

A total of 62 participants (intervention: n = 31; control: n = 31) were randomised and completed the testing within 7 months from the trial start date (February-August 2021), 39 of

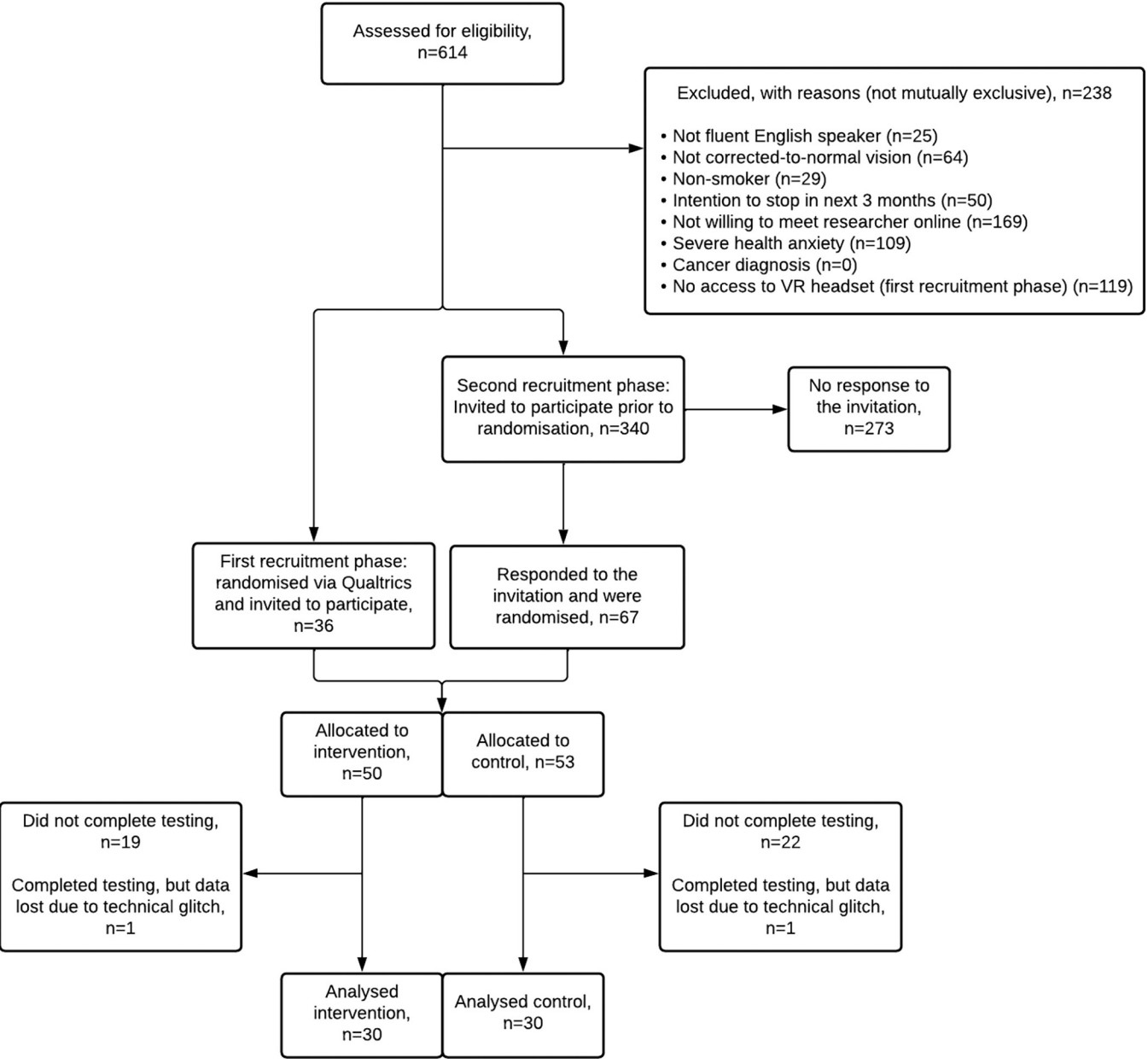

**Fig 3. CONSORT flow diagram of the study participants.**

whom were recruited within 2 months of active recruitment (rather than calendar time) following an amendment to gift inexpensive headsets, sent to participants via post. Due to the need for a protocol amendment, the study was first paused to await ethical approval and next paused for a couple of weeks when the researchers were on leave for the summer holidays. We therefore consider here only the active recruitment periods.

## 3.4 Acceptability

The intervention (86.7%, 95% CI = 69.3%-96.2%) and control (93.3%, 95% CI = 77.9%-99.2%) scenarios were rated as acceptable. In the sensitivity analysis with the acceptability indicators

**Table 1. Participant demographic and smoking characteristics (analytic sample, N = 60).**

| | Total (N = 60) | Intervention (n = 30) | Control (n = 30) |
|---|---|---|---|
| **Age, mean (SD)** | 34.4 (12.1) | 35.9 (13.6) | 32.9 (10.3) |
| **Gender, n (%)** | | | |
| Female | 28 (46.7%) | 17 (56.7%) | 11 (36.7%) |
| Male | 32 (53.3%) | 13 (43.3%) | 19 (63.3%) |
| **Occupational status, n (%)** | | | |
| Manual | 13 (21.7%) | 6 (20.0%) | 7 (23.3%) |
| Non-manual | 30 (50.0%) | 14 (46.7%) | 16 (53.3%) |
| Other (student, unemployed, retired) | 17 (28.3%) | 10 (33.3%) | 7 (23.3%) |
| **Country, n (%)** | | | |
| United Kingdom | 48 (80.0%) | 22 (73.3%) | 26 (86.7%) |
| Austria | 1 (1.7%) | 1 (3.3%) | 0 (0.0%) |
| Canada | 2 (3.3%) | 1 (3.3%) | 1 (3.3%) |
| Greece | 1 (1.7%) | 0 (0.0%) | 1 (3.3%) |
| Hungary | 1 (1.7%) | 0 (0.0%) | 1 (3.3%) |
| India | 1 (1.7%) | 0 (0.0%) | 1 (3.3%) |
| Italy | 2 (3.3%) | 2 (6.7%) | 0 (0.0%) |
| Poland | 1 (1.7%) | 1 (3.3%) | 0 (0.0%) |
| South Africa | 1 (1.7%) | 1 (3.3%) | 0 (0.0%) |
| Spain | 1 (1.7%) | 1 (3.3%) | 0 (0.0%) |
| United States | 1 (1.7%) | 1 (3.3%) | 0 (0%) |
| **Cigarettes per day, mean (SD)** | 9.8 (7.2) | 11.3 (7.6) | 8.4 (6.5) |
| **Time to first cigarette, n (%)** | | | |
| <5 minutes | 7 (11.7%) | 4 (13.3%) | 3 (10.0%) |
| 6–30 minutes | 27 (45.0%) | 15 (50.0%) | 12 (40.0%) |
| 31–60 minutes | 6 (10.0%) | 2 (6.7%) | 4 (13.3%) |
| 60+ minutes | 20 (33.3%) | 9 (30.0%) | 11 (36.7%) |
| **Motivation to stop, n (%)** | | | |
| I don't want to stop smoking | 21 (35.0%) | 9 (30.0%) | 12 (40.0%) |
| I think I should stop smoking but don't really want to | 27 (45.0%) | 16 (53.3%) | 11 (36.7%) |
| I want to stop smoking but haven't thought about when | 8 (13.3%) | 3 (10.0%) | 5 (16.7%) |
| I really want to stop smoking but I don't know when I will | 4 (6.7%) | 2 (6.7%) | 2 (6.7%) |
| **Quit attempt(s), n (%)** | | | |
| No, never | 27 (45.0%) | 9 (30.0%) | 18 (60.0%) |
| Yes, but not in the past year | 26 (43.3%) | 17 (56.7%) | 9 (30.0%) |
| Yes, in the past year | 7 (11.7%) | 4 (13.3%) | 3 (10.0%) |
| **Ever use of behavioural support, n (%)** | | | |
| No | 48 (80.0%) | 22 (73.3%) | 26 (86.7%) |
| Yes | 12 (20.0%) | 8 (26.7%) | 4 (13.3%) |
| **Ever use of pharmacological support, n (%)** | | | |
| No | 32 (53.3%) | 15 (50.0%) | 17 (56.7%) |
| Yes | 28 (46.7%) | 15 (50.0%) | 13 (43.3%) |
| **Own headset, n (%)** | | | |
| No | 31 (51.7%) | 15 (50.0%) | 16 (53.3%) |
| Yes | 29 (48.3%) | 15 (50.0%) | 14 (46.7%) |
| **VR headset type, n (%)** | | | |
| Google Cardboard | 6 (10.0%) | 1 (3.3%) | 5 (16.7%) |
| Homido | 1 (1.7%) | 1 (3.3%) | 0 (0.0%) |

(*Continued*)

**Table 1.** (Continued)

|  | Total (N = 60) | Intervention (n = 30) | Control (n = 30) |
|---|---|---|---|
| Oculus Quest/Rift | 7 (11.7%) | 3 (10.0%) | 4 (13.3%) |
| PlayStation VR | 2 (3.3%) | 0 (0.0%) | 2 (6.7%) |
| Populous | 1 (1.7%) | 0 (0.0%) | 1 (3.3%%) |
| Utopia 360 | 1 (1.7%) | 1 (3.3%) | 0 (0.0%) |
| Trust | 1 (1.7%) | 1 (3.3%) | 0 (0.0%) |
| Don't know/not reported/does not own a headset | 41 (68.3%) | 23 (76.7%) | 18 (60%) |
| **Past VR experience, n (%)** |  |  |  |
| None | 12 (20.0%) | 7 (23.3%) | 5 (16.7%) |
| Limited | 23 (38.3%) | 13 (43.3%) | 10 (33.3%) |
| Some | 19 (31.7%) | 8 (26.7%) | 11 (36.7%) |
| Substantial | 6 (10.0%) | 2 (6.7%) | 4 (13.3%) |

*Note*. SD = standard deviation.

operationalised as continuous variables, there were no significant differences between the intervention and control scenarios (Table A in S1 Table).

## 3.5 Proximal quitting outcomes

Proximal quitting outcomes in the intervention and control arm were comparable (see Table 2). In the sensitivity analysis with the proximal quitting outcomes operationalised as continuous variables (except for intention to stop), there were no significant differences between the intervention and control scenarios, with the exception of quitting self-efficacy (see S1 Table). Quitting self-efficacy was significantly greater in the control (M = 3.03, SD = 1.07) compared with the intervention arm (M = 2.30, SD = 0.95), $p$ = 0.006.

**Table 2.** Proximal quitting outcomes (analytic sample, N = 60).

|  | Intervention (n = 30) | Control (n = 30) |
|---|---|---|
| **Perceived susceptibility to cancer, % (95% CI)** |  |  |
| Susceptible | 93.3% (77.9%-99.2%) | 93.3% (77.9%-99.2%) |
| **Perceived susceptibility to heart disease, % (95% CI)** |  |  |
| Susceptible | 73.3% (54.1%-87.7%) | 86.7% (69.3%-96.2%) |
| **Perceived susceptibility to lung disease, % (95% CI)** |  |  |
| Susceptible | 93.3% (77.9%-99.2%) | 93.3% (77.9%-99.2%) |
| **Cancer response-efficacy, % (95% CI)** |  |  |
| Confident | 80.0% (61.4%-92.3%) | 76.7% (57.7%-90.1%) |
| **Heart disease response-efficacy, % (95% CI)** |  |  |
| Confident | 63.3% (43.9%-80.1%) | 70.0% (50.6%-85.3%) |
| **Lung disease response-efficacy, % (95% CI)** |  |  |
| Confident | 83.3% (65.3%-94.4%) | 86.7% (69.3%-96.2%) |
| **Quitting self-efficacy, % (95% CI)** |  |  |
| Confident | 13.3% (3.7%-30.7%) | 26.7% (12.3%-45.9%) |
| **Behavioural indicator of intention to stop smoking, % (95% CI)** |  |  |
| Yes | 3.3% (0.1%-17.2%) | 0% (0%-11.6%) |

*Note*. CI = confidence interval.

## 4. Discussion

This pilot randomised trial aimed to examine the feasibility of recruitment and acceptability of a brief, theory-informed VR scenario and estimate proximal quitting outcomes. The target sample size of 60 participants was not achieved within the pre-specified feasibility window. However, an amendment to gift inexpensive VR headsets, sent to UK participants via post, appeared promising. This resulted in the recruitment of ~62% of the sample within two months of active recruitment. The intervention and control VR scenarios alike were rated as acceptable by participants. As it may be perceived as sensitive to raise the topic of smoking cessation to smokers unmotivated to stop, we consider the results pertaining to acceptability particularly promising. The perceived susceptibility and response-efficacy to cancer, heart disease, and lung disease were comparably high across arms, which may be indicative of a ceiling effect (i.e., participants already felt moderately to highly susceptible). Quitting self-efficacy and intention to stop smoking were comparably low across arms. There are different plausible interpretations of this finding. First, the brief VR scenario may have been insufficiently immersive or persuasive for boosting smokers' self-efficacy and therefore also their motivation to stop. Second, smokers may have slightly rushed through the post-task survey and hence missed the link with more information about smoking cessation. This merits further exploration in, for example, a study using think aloud methodology, with smokers verbalising their thoughts and impressions whilst viewing the VR scenario.

Prior smoking cessation research using VR technology has primarily focused on the delivery of cue exposure therapy [14], with the exception of a pilot study of a motivational VR scenario for young adults focused on the progression of smoking-related disease [17]. More recently, however, Machulska and colleagues conducted an RCT of a VR-based approach bias retraining programme, with no significant effects on approach bias or daily cigarette consumption detected at a 7-week follow-up [35]. Since the planning and conduct of the present study, VR technology has been used to deliver educational games to high school students, with a view to preventing smoking initiation [36]. In addition, VR technology has been used to deliver post-retrieval extinction therapy to disrupt smoking memory reconsolidation and prevent future cravings, with positive preliminary results [37]. However, large-scale trials of VR deployments for smoking cessation with long-term follow-ups (e.g., 6 or 12 months) remain scarce.

### 4.1 Strengths and limitations

This study was strengthened by including the element of randomisation, with participants allocated to view either an active or a 'sham' VR scenario. In addition, the VR scenarios were delivered in smokers' own homes, thus mimicking real-world conditions, and ensuring ecological validity of the procedures and results.

However, this study also had several limitations. First, as illustrated by the initially slow recruitment, smokers do not yet appear to have easy access to VR headsets. Until VR headsets are more ubiquitous among the general population of smokers, the potential for VR to deliver motivational messaging in smokers' own homes is limited. In addition, as most of our sample had at least some prior VR experience, this raises the question as to whether an immersive VR scenario focused on the health consequences of smoking may be more salient to those using VR for the first time (i.e., a potential novelty effect) [38,39]. Second, the effective point of randomisation occurred when participants booked a meeting to complete the online testing (with intervention and control participants receiving the same instructions), with many participants dropping out at this stage (i.e., not attending the testing). Although not statistically significant, this led to group imbalances in the baseline sociodemographic and smoking characteristics.

Although this did not compromise the randomisation itself (as participants had received identical instructions, with drop-outs unrelated to group allocation), future studies should aim to randomise participants as close to participation as possible (e.g., when attending the testing session). Third, our active VR scenario was not highly interactive (i.e., albeit immersive, smokers were unable to communicate with the chest physician through a joystick or similar tools). This may have negatively influenced feelings of immersion and hence also smokers' affective responses to the scenario, which may in turn have been insufficient for boosting their motivation to stop [8]. However, the findings pertaining to the proximal quitting outcomes ought to be interpreted with caution, as the study was not powered to detect group differences in smokers' perceptions and intentions. Fourth, to ensure that participants completed the task as instructed, a researcher was present during the testing via teleconferencing software. This may have influenced participants' willingness to click on the link with additional information about smoking cessation (which was used to capture their intention to stop)–i.e., they may have slightly rushed through the post-task survey. Fifth, the focus on proximal quitting outcomes (e.g., intention to stop) is limited by the well-known intention-behaviour gap (i.e., strong intentions often do not translate to actual behaviour change) [40]. Therefore, future work would benefit from capturing quit attempts and quit success in addition to perceptions and intentions. Sixth, the items used to capture acceptability and intention to stop smoking were developed for the purposes of the present study and had not been validated. Though, we have previously used a similar behavioural indicator of intention to change in a different behaviour change context [41] and we note that prior research has found a positive association between intention to change and information seeking/user engagement in digital health research [42]. With regards to the definition and operationalisation of intervention acceptability, this remains a debated topic and we have previously argued that acceptability can usefully be conceptualised as an emergent property of a complex, adaptive system of interacting sub-components (e.g., cognitive and affective attitude) [20]. However, it should be noted that there may be some degree of overlap between acceptability as operationalised in the present study (e.g., anger, distress, perceived usefulness) and the emotional reactions expected to be elicited by the VR scenario (e.g., fear, worry)–which in turn were expected to influence perceived susceptibility to cancer/heart disease/lung disease. Although correlations between the acceptability and susceptibility indicators were small in magnitude and non-significant (presented in Table B in S1 Table), further conceptual and psychometric work is required to refine the acceptability measure and evaluate its validity and reliability (including the identification of a suitable cut-off score across the three indicators). In addition, future research would benefit from assessing the prospective validity of behavioural indicators of intention (i.e., whether website clicks are associated with self-reported smoking cessation attempts measured at a later time point). Finally, due to a coding error when setting up the Qualtrics survey, the validated scale used to capture response-efficacy was measured using seven (instead of five) response options.

## 4.2 Implications and avenues for future research

Until VR headsets become more widespread among the general population of smokers, providing smokers with a simple headset (<£10) appears promising. Alternatively, the opportunistic delivery of immersive stop smoking messaging in settings other than smokers' homes (e.g., general practice or dentist waiting rooms) merits further investigation. In addition, prior to conducting a larger-scale study, we recommend optimising the VR scenario through drawing on user-centred design principles, identifying ways to make the active VR scenario more interactive and immersive, and better understanding smokers' experiences of an optimised

scenario (including the motivational message itself) through think aloud methodology [43–45]. Areas of improvement may, for example, include the addition of functionality to allow users to directly interact/communicate with the physician (e.g., through selecting different communication options via a joystick). Future work would also benefit from formalising some of the insights gleaned about users' experiences from informal conversations with participants at the end of testing sessions through qualitative methods.

### 4.3 Conclusions

This was, to our knowledge, the first pilot randomised trial of a VR scenario designed specifically for adult smokers unmotivated to stop. The target sample size was not achieved within the feasibility window; however, an amendment to gift inexpensive VR headsets appeared promising. The VR scenario appeared acceptable. Future work would benefit from optimising the VR scenario prior to conducting a larger-scale study.

## Supporting information

**S1 CONSORT Checklist. CONSORT checklist of information to include when reporting a pilot or feasibility trial.**
(DOC)

**S1 Data. Behaviour change techniques (BCTs) included in the intervention scenario, coded against a 44-item taxonomy of BCTs used in behavioural smoking cessation interventions [33].**
(XLSX)

**S1 Table. Results from linear regression analyses with the outcome variables (except for intention to stop smoking) operationalised as continuous in addition to the bivariate correlations between the acceptability and perceived susceptibility indicators.**
(DOCX)

## Acknowledgments

We gratefully acknowledge the funding listed above. The researchers would like to thank Nick Abelson from MasterChange and Dr. Matt Evison for providing access to the active VR scenario and HybridMedical for providing access to the control scenario.

## Author Contributions

**Conceptualization:** Olga Perski, Trupti Jambharunkar, Jamie Brown, Dimitra Kale.

**Data curation:** Olga Perski, Trupti Jambharunkar, Dimitra Kale.

**Formal analysis:** Olga Perski.

**Funding acquisition:** Jamie Brown.

**Investigation:** Olga Perski.

**Methodology:** Olga Perski, Jamie Brown, Dimitra Kale.

**Project administration:** Trupti Jambharunkar, Dimitra Kale.

**Writing – original draft:** Olga Perski.

**Writing – review & editing:** Olga Perski, Trupti Jambharunkar, Jamie Brown, Dimitra Kale.

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
