## [Decision Letter · Decision Letter 0]

10 Feb 2022

PDIG-D-22-00003

A pilot randomised trial of a brief virtual reality scenario in smokers unmotivated to quit: Assessing the feasibility of recruitment

PLOS Digital Health

Dear Dr. Perski,

Thank you for submitting your manuscript to PLOS Digital Health. After careful consideration, we feel that it has merit but does not fully meet PLOS Digital Health's publication criteria as it currently stands. Therefore, we invite you to submit a revised version of the manuscript that addresses the points raised during the review process.

The reviewers see potential in your submission, however, they also highlighted a number of issues including:

- a lack of detial in describing immersive VR and the intervention, including behaviour change techniques used

- the decision to dichotomise variables in the analysis

- the discussion being short and not referring at all to research question 3.

We look forward to receiving your revised manuscript.

Kind regards,

Laura M. König

Academic Editor

PLOS Digital Health

Journal Requirements:

1. Please amend your detailed Financial Disclosure statement. This is published with the article, therefore should be completed in full sentences and contain the exact wording you wish to be published.

State what role the funders took in the study. If the funders had no role in your study, please state: “The funders had no role in study design, data collection and analysis, decision to publish, or preparation of the manuscript.”

2. Please update the completed 'Competing Interests' statement. Please declare all competing interests beginning with the statement "I have read the journal's policy and the authors of this manuscript have the following competing interests:"

3. Please provide separate figure files in .tif or .eps format only and remove any figures embedded in your manuscript file. Please ensure that all files are under our size limit of 20MB.

For more information about how to convert your figure files please see our guidelines: https://journals.plos.org/digitalhealth/s/figures

4. We have noticed that you have uploaded supporting information but you have not included a list of legends. Please add a full list of legends for all supporting information files (including figures, table and data files) after the references list.

Additional Editor Comments (if provided):

Reviewers' comments:

Reviewer's Responses to Questions

**Comments to the Author**

1. Does this manuscript meet PLOS Digital Health’s publication criteria? Is the manuscript technically sound, and do the data support the conclusions? The manuscript must describe methodologically and ethically rigorous research with conclusions that are appropriately drawn based on the data presented.

Reviewer #1: Yes

Reviewer #2: Partly

Reviewer #3: Yes

2. Has the statistical analysis been performed appropriately and rigorously?

Reviewer #1: Yes

Reviewer #2: No

Reviewer #3: Yes

3. Have the authors made all data underlying the findings in their manuscript fully available (please refer to the Data Availability Statement at the start of the manuscript PDF file)?

Reviewer #1: Yes

Reviewer #2: Yes

Reviewer #3: Yes

4. Is the manuscript presented in an intelligible fashion and written in standard English?

Reviewer #1: Yes

Reviewer #2: Yes

Reviewer #3: Yes

5. Review Comments to the Author

Reviewer #1: Thanks for the opportunity to review this interesting manuscript about a novel VR-based strategy to reach and help smokers unmotivated to quit. The manuscript is easy to read and generally offers a sound overview of the research that was conducted. I would advise the authors to take a look at the feedback points below, to make further improvements to their manuscript.

- the introduction is well-written and offers a clear overview of the main concepts and rationale for main choices. At several points, the introduction raises questions, but these are always answered when one continues reading the next sections. So, overall the introduction is on point and well-substantiated with references.

- the methods section includes a rather detailed account of how the pilot trial was set up, how inclusion and recruitment took place, and what data was collected in both study arms. Yet, the section describing the intervention conditions could elaborate more on the specific working mechanism of the immersive VR component (e.g. in comparison to watching a video of a physician talking to you); what makes VR potentially more effective than other kind of media outlets/channels?

- although the results and conclusions seem clear, it would be interesting to read more about potential improvements to the active VR scenario. According to the authors, what may be the most promising application of VR for smoking cessation, and which target population should this be particularly aimed at?

Minor comments

- there is some redundancy in the information provided in the second paragraph of section 2.4 measures and procedures.

- the part describing and justifying the recruitment goal is somewhat confusing: could you make it more explicit how the 60 participants in 3 months goal resulted from the full-trial calculations provided?

- results: could you clarify why only 103 out of 376 eligible participants were randomized?

- results: it is not clear what is meant with the following sentence '39 of whom were recruited within 2 months of active recruitment (rather than calendar time) following an amendment to gift inexpensive headsets, sent to participants via post.' Please consider rephrasing.

Reviewer #2: This study sets out to examine the feasibility and acceptability of a pilot RCT using a brief virtual reality (VR) intervention in 60 smokers unmotivated to stop, randomized to an intervention (motivational VR) or control condition (control VR). Results showed that the targeted sample size was not achieved within the planned time frame, but that the intervention was perceived as acceptable. Preliminary analysis of the intervention effect on motivational constructs showed comparable point estimates in perceived susceptibility, response-efficacy and self-efficacy, and behavioral intention.

Overall, this paper takes an interesting approach using VR to deliver motivational messages. The manuscript is well structured and written and the experimental design is a strength. I however also have several concerns that narrow the impact of the manuscript. Overall, I find that in several instances the paper is rather short on information and appears superficial. I am also struggling that the sample recruitment is treated as the primary outcome and substantive questions which most likely are more relevant for the broader scientific community, are moved into the background. Also, there are several methodological concerns concerning the validity of measures and analysis (e.g., dichotomization). Please see more specific comments below:

Introduction:

- Please elaborate on EPPM and what the theory-basis for the intervention scenario is.

- It is unclear if authors are interested in the acceptability of VR per se or the specific content delivered via VR in the study. 

- Please elaborate on the concept of acceptability of digital health interventions as mutlifaceted concept and provide a rationale for why this is operationalized as attitudes. 

Methods:

- Please provide citations for where the items stem from and provide reliability coeffients along with descriptive data. 

- Why were all measures dichotomized and on what basis? This is in my view highly problematic, resulting in a loss of information, and endangering validity. 

- How valid is the behavioral measure of intention? I am not convinced that clicking on additional evidence-based information reflects intention to stop. It is also plausible that smokers who have 

been motivated might not think they need further information to make a decision. 

- Please elaborate the presented rationale for using point estimates (%). Why was it not possible to conduct regression analyses and report mean level differences and and corresponding CI (instead of P values)? 

Discussion:

- There is no discussion of Research Question 3

Minor points:

p. 5: explain why cancer patients were not eligible to take part in the study

p. 11: What is meant by calendar time?

Reviewer #3: This paper provides a clear report of the pilot RCT conducted by the authors. I have a number of minor comments that could be considered before the article is suitable for publication.

p. 3: suggest specifying what exactly is meant by an immersive VR scenario at this stage in the article (i.e., what sensory substitutions are necessary to facilitate this experience – vision, hearing, touch – and what equipment is absolutely necessary for the user to access the intervention). A more technical definition of immersive VR in the context of this study would also be more informative, offering a greater understanding of what the end-user is experiencing.

p. 3: further description of the specific features (presence/co-presence, immersion) of immersive VR that are being utilized in this intervention would be important here due to the novelty of the technology – particularly in this field. Linked to this, as immersion appears to be a key hypothesised determinant of research question 3, further discussion about immersion as a phenomenon would strengthen the rationale for exploring the use of immersive VR in this context.

p. 8: suggest including any data available on the user testing conducted in December 2020 (researcher notes/observations, reflexive notes, etc.) as it would offer more insight into the processes involved in designing this intervention as suggested in the recommendations offered at the end of this article – further qualitative research into end-users’ experiences and the application of user-centred design principles. 

p. 12: if the data is available, it would be useful to know what other immersive VR devices participants who already had the equipment were using. Current immersive VR technologies can differ significantly, affecting key experiences such as presence and immersion.

p. 13: suggest supporting claims of novelty effects and the effects varying levels of immersion have on immersive VR participants’ experiences with past research. 

p. 13: typological error – “third” used twice when outlining limitations

6. PLOS authors have the option to publish the peer review history of their article (what does this mean?). If published, this will include your full peer review and any attached files.

**Do you want your identity to be public for this peer review?** For information about this choice, including consent withdrawal, please see our Privacy Policy.

Reviewer #1: Yes: Dennis de Ruijter

Reviewer #2: No

Reviewer #3: No

---

## [Decision Letter · Decision Letter 1]

20 Apr 2022

PDIG-D-22-00003R1

A pilot randomised trial of a brief virtual reality scenario in smokers unmotivated to quit: Assessing the feasibility of recruitment

PLOS Digital Health

Dear Dr. Perski,

Thank you for submitting your manuscript to PLOS Digital Health. After careful consideration, we feel that it has merit but does not fully meet PLOS Digital Health's publication criteria as it currently stands. Therefore, we invite you to submit a revised version of the manuscript that addresses the points raised during the review process.

Based on additional comments submitted by one of the reviewers, I would like to ask the authors to make the following changes:

- Briefly summarize the results presented in the supplementary material in the main text to contextualize the meaning of "remained similar".

- The authors state that the concept of acceptability is debated; this debate should be reflected in the discussion.

- The reviewer highlights that acceptability and susceptibility might be strongly correlated due to the wording of the items. This is a valid concern should be addressed.

We look forward to receiving your revised manuscript.

Kind regards,

Laura M. König

Academic Editor

PLOS Digital Health

Journal Requirements:

Additional Editor Comments (if provided):

Reviewers' comments:

Reviewer's Responses to Questions

**Comments to the Author**

1. If the authors have adequately addressed your comments raised in a previous round of review and you feel that this manuscript is now acceptable for publication, you may indicate that here to bypass the “Comments to the Author” section, enter your conflict of interest statement in the “Confidential to Editor” section, and submit your "Accept" recommendation.

Reviewer #1: All comments have been addressed

Reviewer #2: (No Response)

Reviewer #3: All comments have been addressed

2. Does this manuscript meet PLOS Digital Health’s publication criteria? Is the manuscript technically sound, and do the data support the conclusions? The manuscript must describe methodologically and ethically rigorous research with conclusions that are appropriately drawn based on the data presented.

Reviewer #1: Yes

Reviewer #2: Partly

Reviewer #3: Yes

3. Has the statistical analysis been performed appropriately and rigorously?

Reviewer #1: Yes

Reviewer #2: No

Reviewer #3: Yes

4. Have the authors made all data underlying the findings in their manuscript fully available (please refer to the Data Availability Statement at the start of the manuscript PDF file)?

Reviewer #1: Yes

Reviewer #2: Yes

Reviewer #3: Yes

5. Is the manuscript presented in an intelligible fashion and written in standard English?

Reviewer #1: Yes

Reviewer #2: Yes

Reviewer #3: Yes

6. Review Comments to the Author

Reviewer #1: I would like to thank the authors for carefully addressing all the feedback received. I do not have any additional remarks. Good luck with the publication process!

Reviewer #2: I would like to thank the authors for taking the time and effort to revise their manuscript. I appreciate the clarifications throughout the manuscript, but still have a few questions or concerns, mainly regarding to the operationalization and validity of measures:

1. I am not fully convinced that the benefit of the dichotomization in aiding the interpretation of results outweighs the concern of loss of information. The concepts measured clearly reflect continuous concepts and a clear rationale for the cutoffs chosen is missing. From my personal viewpoint, the previously planned analyses with regressions (maybe except for intention to change) would yield more informative results. 

2. As to the additional analyses in Supplementary File 3, can the authors explain what “remain similar” means? Again, I belief that reporting (descriptive) means intervention vs. control would be more informative than proportions across the likert scale. 

3. I am still struggling to grasp as to how the affective beliefs should capture acceptability. One item example is “made me feel distressed”, and if participants report that the disagreed this was coded as acceptable. However, the scenario was expected to raise emotions, so distress might reflect that health consequences were perceived as intended in the scenario (similar to a manipulation check)? This might pose a problem for the measure of acceptability. Can the authors please explain how this should conceptualize non-acceptability? Also, was a mean score of cognitive and affective attitudes calculated? If yes, please explicitly state so.

4. What is the correlation among the variables assessed. As with regard to my second concern, I am wondering if acceptability might correlate/overlap strongly with the measure of susceptibility.

Reviewer #3: (No Response)

7. PLOS authors have the option to publish the peer review history of their article (what does this mean?). If published, this will include your full peer review and any attached files.

**Do you want your identity to be public for this peer review?** For information about this choice, including consent withdrawal, please see our Privacy Policy.

Reviewer #1: No

Reviewer #2: No

Reviewer #3: Yes: David Healy

---

## [Editor Report · Decision Letter 2]

7 May 2022

A pilot randomised trial of a brief virtual reality scenario in smokers unmotivated to quit: Assessing the feasibility of recruitment

PDIG-D-22-00003R2

Dear Dr Perski,

We are pleased to inform you that your manuscript 'A pilot randomised trial of a brief virtual reality scenario in smokers unmotivated to quit: Assessing the feasibility of recruitment' has been provisionally accepted for publication in PLOS Digital Health.

Best regards,

Laura M. König

Academic Editor

PLOS Digital Health